# Effect of a Selected Protective Culture of *Lactilactobacillus sakei* on the Evolution of Volatile Compounds and on the Final Sensorial Characteristics of Traditional Dry-Cured Fermented “Salchichón”

**DOI:** 10.3390/biology12010088

**Published:** 2023-01-05

**Authors:** Irene Martín, Carmen García, Alicia Rodríguez, Juan J. Córdoba

**Affiliations:** 1Higiene y Seguridad Alimentaria, Instituto Universitario de Investigación de Carne y Productos Cárnicos (IProCar), Universidad de Extremadura, Avda. de las Ciencias, s/n., 10003 Cáceres, Spain; 2Tecnología y Calidad de Alimentos, Instituto Universitario de Investigación de Carne y Productos Cárnicos (IProCar), Universidad de Extremadura, Avda. de las Ciencias, s/n., 10003 Cáceres, Spain

**Keywords:** lactic acid bacteria, dry-cured fermented sausages, sensorial characteristics

## Abstract

**Simple Summary:**

The use of lactic acid bacteria as starter culture in the production of traditional dry-cured fermented sausages is not widespread due to the possible undesirable effects of these bacteria on the sensory quality of the product. In this work, it is demonstrated that the use of a selected *Lactilactobacillus sakei* as protective culture in traditional “salchichón” does not negatively affect the sensory parameters including texture and color and could even improve the flavor of this traditional product.

**Abstract:**

Background: In this work, the effect of a selected starter culture of *Lactilactobacillus sakei* 205 on the evolution of volatile compounds throughout the ripening process and on the final sensorial characteristics of traditional dry-cured fermented “salchichón” was evaluated. Methods: “Salchichón” sausages were prepared, inoculated with *L. sakei* 205, and ripened for 90 days. Volatile compounds were analyzed throughout the ripening by GC-MS. In the final product, instrumental texture and color were determined. In addition, sensorial analysis was performed by a semi-trained panel. Results: The inoculation of *L. sakei* 205 does not influence the texture and color parameters of ripened “salchichón”. However, an increase in volatile compounds derived from amino acid catabolism and microbial esterification and a decrease in compounds derived from lipid oxidation, mainly hexanal, were observed throughout the ripening time as a consequence of *L. sakei* inoculation, which could have a positive effect on the flavor development of the dry-cured fermented “salchichón”. Conclusions: The use of selected strains of lactic acid bacteria (LAB) such as *L. sakei* 205 as a protective culture could be recommended to improve the quality of traditional “salchichón”.

## 1. Introduction

Traditional Iberian dry-cured fermented “salchichón” sausage is a high-value product obtained from autochthonous Iberian pigs in Spain. Briefly, the production of this meat product consists of first comminuting and blending pork meat and fat (4 °C), and the addition of spices, carbohydrates, and salt (NaCl). Next, casings are filled under vacuum, and the obtained sausages are dried and ripened in drying rooms at temperatures increasing from 5 to 12 °C and relative humidities decreasing from 85 to 70%. In these conditions, the sausages must be maintained between 1 and 3 months (depending on the type and diameter of the casing used), to allow the water activity (a_w_) to slowly decrease below 0.92 a_w_. The specific requirements differ in the specialty of the technology typical to a region or country [1]. Iberian dry-cured fermented sausages are usually elaborated following traditional recipes without using starter cultures and are ripened for at least 2–3 months to allow the formation of characteristic flavor. Since this type of product has been spontaneously fermented depending on the natural microbial population, the flavor and texture of the final product are difficult to guarantee, as it has been reported in meat products of similar characteristics [2,3]. This problem could be solved by the addition of safe protective cultures, but this use is not well extended to this product. However, its utilization has provided good results in relation to taste and texture improvement in other foodstuffs such as cheeses [4]. 

Indigenous LAB, belonging to dry-cured fermented sausages, have been reported as microorganisms that are especially well adapted to the ecological conditions of meat fermentation which control the ripening processes and inhibit the growth of spontaneous microorganisms including *Listeria monocytogenes,* the most worrying microorganism in this kind of product [5,6]. The *Lactilactobacillus sakei* 205 strain isolated from traditional Iberian dry-cured fermented sausages has been selected by its adaptation to the ripening process of these products and its capacity to control the growth of undesirable microorganisms in Iberian dry-cured fermented sausages (spoilage and pathogen microorganisms) [7]. In addition, this LAB species has the recognition of Generally Recognized as Safe (GRAS) according to the U.S. Food and Drug Administration (FDA) and the Qualified Presumption of Safety (QPS) from the European Food Safety Authority (EFSA) [8]. Furthermore, the addition of selected LAB species such as *L. sakei* can ensure that the typical flavor of “salchichón” does not change, even if this product is elaborated elsewhere. However, before proposing the use of *L. sakei* 205 as a protective culture in the elaboration of traditional Iberian dry-cured fermented “salchichón”, the evaluation of its effect on the evolution of volatile compounds involved in flavor development throughout “salchichón” ripening and on the sensorial characteristics of the final product is necessary. This evaluation should be carried out in real conditions of processing, taking into account that in addition to the selected *L. sakei* strain, other origin microorganisms, mainly indigenous LAB, could be present in the product throughout the ripening. 

This study aimed to evaluate the effect of the selected strain *L. sakei* 205 on the evolution of volatile compounds throughout the ripening process and on the sensorial characteristics of traditional Iberian dry-cured fermented “salchichón” to achieve more consistent flavor and taste, and a uniform appearance. 

## 2. Materials and Methods

### 2.1. Lactic Acid Bacterium Culture

The strain *L. sakei* 205 was isolated from Iberian dry-cured fermented sausages and selected by its anti-*L. monocytogenes* activity in basic culture medium and in real “salchichón” samples [7]. The inoculum of LAB was obtained by inoculating 100 µL of a culture preserved at −80 °C on Man Rogosa Sharpe (MRS) broth (Fisher Bioreagents, Geel, Belgium) and incubating it at 30 °C for 48 h. Later, a volume of 100 µL of the inoculum was plated on an MRS agar plate (Oxoid, UK) plate and incubated again at 30 °C for 48 h, and the counts of the microorganism were conducted. Once the concentration of cells was known (approximately 8–9 log CFU/mL), decimal dilutions were made to achieve the concentration of 7 log CFU/mL which was used as inoculum according to the method described by Martín et al. [7]. 

### 2.2. Preparation of Iberian Dry-Cured Fermented “Salchichón” Sausages

Two equal batches of Iberian dry-cured fermented “salchichón” sausages were manufactured: a non-inoculated batch (Control) and another batch inoculated with *L. sakei* 205 at 7 log CFU/g (Ls). The inoculum of *L. sakei* was adjusted and prepared to be resuspended in a total volume of 150 mL of Phosphate Buffered Saline (PBS). In the case of the Control batch, a volume of 150 mL of sterilized PBS was added instead of the bacterium inoculum. 

Meat dough used for the manufacture of “salchichón” was made in a traditional meat industry in Spain and it consisted of Iberian pork meat and Iberian pig fatback, salt, sugar, potassium nitrate, sodium nitrite, black pepper, and spices. Next, after inoculation of the mixtures of each batch (PBS in Control batch and *L. sakei* 205 in Ls batch), they were stuffed into regenerated collagen casings (40 mm in diameter) and ripened in the Research Pilot Plant facility at the Faculty of Veterinary Science, University of Extremadura (Cáceres, Spain) for 90 days, adjusting the relative humidity (RH) and temperature to traditional processing conditions in the meat industry for this type of product. Thus, the ripening started with 3 days at 5 °C and 85% RH, followed by 17 days at 7 °C and 80% RH, 10 days at 9 °C and 75% RH, and, finally, the sausages were kept at 12 °C and 70% RH to reach 90 days of ripening [7]. 

The texture, color, and sensory analysis were only determined at the end of the processing (90 days). However, the analysis of the volatile compounds was determined at 0, 30, 60, and 90 days of ripening. For all the analysis, five sausages were taken at each sampling time. Thus, all the tests were carried out in quintuplicate. 

The levels of LAB on the “salchichón” after stuffing (day 0) and throughout the ripening were tested in MRS agar after incubating plates under microaerophilic conditions at 30 °C for 48 h. In the inoculated batch, counts of LAB were always higher than 6 log CFU/g throughout the processing. In the Control batch, LAB values at the beginning of the ripening process were lower than 4 log CFU/g, but during processing, they reached levels higher than 6 log CFU/g. In addition, in the inoculated *L. sakei* batch, the implantation of *L. sakei* 205 was tested at the end of ripening (90 days). For this, 50% of the characteristic LAB colonies of MRS plates were randomly tested by sequencing analysis of 16S rRNA, and PFGE analysis by using the restriction *NotI* and *SgsI* enzymes (Thermo Fisher Scientific, USA) [7]. Most of the 85% of the investigated LAB isolates were identified as *L. sakei* by sequencing analysis. These isolates showed the same pattern as *L. sakei* 205 in the PFGE analysis. Thus, *L. sakei* 205 was well implanted in the dry-cured fermented sausages throughout the ripening.

### 2.3. Volatile Compound Analysis

The volatile compounds in dry-cured sausages “salchichón” were extracted by solid-phase microextraction (SPME) after heating to 37 °C for 30 min, using a divinylbenzene-carboxen-polydimethylsiloxane (DVB/CAR/PDMS) 50/30 µm fiber (Merck; Darmstadt, Germany). They were then analyzed by gas chromatography–mass spectrometry (GC-MS) in a Gas Chromatograph 6890 GC (Agilent Technologies; Santa Clara, CA, USA) equipped with an HP-5 column (5% phenyl–95% dimethylpolysiloxane) and coupled to a mass spectrometer (MS) detector, 5975C (Agilent Technologies). The oven temperature started at 40 °C for 5 min and was increased to 280 °C, with a rate of 7 °C/min. The desorption time was 30 min at 250 °C. The transfer line temperature was established at 280 °C. The carrier gas was helium with a flow rate of 1.2 mL/min. MS detection was performed in full scan (50–350 amu). Automated peak search and spectral deconvolution were used for data treatment, and the identification of the volatile compounds was achieved by comparing their mass spectra with the NIST/EPA/NIH library. The volatile compound analysis was performed in quintuplicate.

### 2.4. Instrumental Texture

The texture analysis was performed at room temperature using a Texture Profile Analysis (TPA). The instrument used was a TA XT Plus Texture Analyzer (StableMicro Systems Ltd., Godalming, UK) equipped with a cylindrical probe of 5 cm in diameter. This procedure involved cutting slices approximately 1 cm thick. Hardness (N), springiness (cm), cohesiveness, gumminess (N), chewiness (N cm), and adhesiveness (N s) were evaluated at the end of the ripening process [9]. The texture analysis of each quintuplicate was carried out in triplicate.

### 2.5. Instrumental Color

Color was determined on the cut surface of each sample using a Minolta CR-300 colorimeter (Konica Minolta, Inc.; Nieuwegein, The Netherlands) with an illuminant D65, a 0° standard observer and one port/display area of 2.5 cm that was calibrated before use with a white tile having the following values: L* = 93.5, a* = 1.0 and b* = 0.8. Color was expressed according to the Commission International de l’Eclairage (CIE) system and reported as CIE L* (lightness), CIE a* (redness), and CIE b* (yellowness), in which the chroma and hue angle were calculated as (a*^2^ + b*^2^)^0.5^ and tan^−1^ (b*/a*), respectively [10].

### 2.6. Sensory Evaluation

A triangular sensory analysis was carried out in this study with a semi-trained panel (24), including students and lecturers at the Faculty of Veterinary Sciences (University of Extremadura, Caceres, Spain). This study was carried out following the reported guidelines [11] for triangular tests and with the permission of the ethics committee of the University of Extremadura.

Three samples were presented to each panelist member, marked with random three digit codes, and served at room temperature on white plates, and they rated which sample is different. Cookies (with no added salt) and about 200 mL of water were also provided to the panelists to rinse between samples. The panel sessions were held around 2 h before lunch in the sensory panel booth room of the Faculty of Veterinary Sciences of the University of Extremadura in Caceres (Spain). Data about sex and age were also collected. 

### 2.7. Statistical Analyses

The statistical treatment was carried out using SPSS IBM v.22 software (IBM, Armonk, NY, USA). Once the dependent (texture, color, and volatile compounds) and independent (days of ripening and different batches) variables of the analysis were determined, a study of the normality of the different data populations was carried out using the Shapiro–Wilk test. As the results showed a non-parametric distribution the Kruskal–Wallis and Mann–Whitney tests were performed. Statistical significance was established at *p* ≤ 0.05.

## 3. Results and Discussion

### 3.1. Evolution of Volatile Compounds of “Salchichón” throughout the Ripening

A total of 52 volatile compounds were identified throughout the ripening time in the two different batches of “salchichón” (Table 1 and Table 2). Most of these compounds have previously been reported in different types of dry-cured fermented sausages [12,13,14]. These volatile compounds were classified according to their most probable origin as from lipid oxidation (25), microbial esterification (3), carbohydrate fermentation (1), amino acid catabolism (6), and spices (13), and exposed as the summation of the accumulated area of these groups in Figure 1. However, some of the compounds could have more than one source or they are derived from secondary reactions between substances generated in different catabolic routes [15]. When the accumulated area of these groups of compounds was analyzed, it was observed that levels of those derived from lipid oxidation showed a decrease throughout the ripening time in both Control and inoculated batches in comparison with day 0 levels (Figure 1). However, the accumulated area of compounds derived from amino acid catabolism and spices increased throughout the ripening time in the Ls batch, showing the highest mean values at day 60 (derived from amino acid catabolism) and at day 90 of ripening (derived from species). Compounds derived from microbial esterification showed the highest levels in both analyzed batches at day 90 of ripening (Figure 1). 

When specific volatile compounds in each group were analyzed, it was observed that most of the significant (*p* ≤ 0.05) differences were detected between days of ripening in each Control and Ls batch (Table 1 and Table 2), and there were only small differences between batches at each ripening day. Thus, in the group of **compounds derived from lipid oxidation,** the significant (*p* ≤ 0.05) decrease in hexanal, heptanal, and 1-pentanol for most of the ripening time is remarkable. However, there was a significant (*p* ≤ 0.05) increase in 2-octanone in the Control batch during ripening, while it remained stable in the Ls batch. An increase in octanoic acid at day 90 of ripening was found in both Control and Ls batches. There were no consistent differences between Control and Ls batches in volatile compounds derived from lipid oxidation. Linear aldehydes such as hexanal have been reported as the most important aldehyde in dry-cured fermented sausages [16,17], and it has also been chosen as an index of the level of oxidation [18]. Given that the aldehydes, hexanal and heptanal, and the alcohol 1-pentanol are formed by the oxidation of unsaturated fatty acids, their reduction during ripening could be related to the microbial catalase activity in Ls and Control batches, that suppose a retard in autoxidation due to decomposition of hydrogen peroxide [19]. The excessive formation of hexanal gives the dry-cured fermented meat products a flavor of being rancid, pungent, and toasty [20,21]. Thus, the reduction observed for this compound in both batches throughout the ripening process is very interesting for the flavor development of the final product since it may contribute to reducing the note of rancidity of ripened “salchichón”. The increases observed during ripening in 2-octanone may be derived from the oxidation of aldehydes from linoleic acid during the ripening time [19,22].

**Compounds derived from microbial esterification** also showed higher levels at days 60 and 90 of ripening than at the beginning of maturation (Table 1). In addition, in the Ls batch, all these compounds, except hexanoic acid ethyl ester, were detected on all days of ripening while in the Control batch, most of them were only detected at days 60 and 90 of ripening. Ethyl esters, usually present in dry-cured fermented sausages, may arise from the action of LAB inoculated in the Ls batch but also from autochthonous LAB in the Control batch. This could be justified since the compounds from microbial esterification were detected throughout the ripening in the Ls batch, where levels of LAB were always higher than 6 log CFU/g, while in the Control batch were detected mainly at the end of ripening (Figure 1) when this microbial group reached levels higher than 6 log CFU/g. Ethyl esters contribute to a proper dry-cured fermented sausage flavor due to their characteristic fruity notes, their low odor threshold values, and their contribution to masking rancid odors [14,23]. Their presence, together with 3-methylbutanal, has been associated with a ripened flavor [12]. 

**Compounds derived from amino acid catabolism** showed higher levels at days 60 and 90 than at days 0 and 30 of ripening (Table 2, Figure 1), probably due to the increase in the proteolytic activity during ripening, since these compounds were derived mostly from amino acids by Strecker degradation and/or microbial metabolism [24,25,26]. In both analyzed batches, most of these compounds were detected only at days 60 or 90 of ripening, although in the Ls batch, some compounds such as 2-methyl-1-propanol and 3-methyl-butanal were also detected at the beginning of ripening. From all of these compounds, the branched aldehyde 3-methyl-butanal has been reported to contribute considerably to the overall flavor of dry-cured fermented sausages [12,27]. The 2-methyl-propanoic acid and 3-methyl-butanoic acid which were detected at the end of ripening in Ls and Control batches have been reported as compounds derived from the microbial metabolism of Val and Leu, respectively [28]. These compounds have been attributed to a characteristic “sweet” odor [29] and may also have a positive impact on the aroma of dry-cured fermented sausages due to their conversion into fruity esters [23]. Thus, *L. sakei* 205 could contribute to the generation of the compounds derived from amino acid catabolism although this effect is not very relevant when comparing the Ls batch with the Control batch, since, in the latest batch, autochthonous LAB are present at similar counts to *L. sakei* 205.

Regarding **compounds derived from carbohydrate fermentation,** only acetic acid was found (Table 2). This compound was detected on all days of ripening in the Ls batch, and only on some of the days of ripening in the Control batch. Acetic acid mainly derives from the microbial fermentation of sugars from LAB or the Maillard reaction [12]. No significant differences between Control and Ls batches in the levels of this compound were detected, likely due to the inoculation of *L. sakei* 205 which did not suppose an increase in acidity of “salchichón”. 

Most of the volatile **compounds from the added spices** were detected in both analyzed batches of “salchichón” on all days of ripening (Table 2). Terpenes were most of the compounds derived from spices, D-limonene, caryophyllene and 3-carene being predominant in both batches. Particularly, α and β- pinene, D-limonene, and carene [30] were identified in the essential oil of black pepper which is an ingredient added to raw meat and fat during the mixing process. However, the presence of terpenes has no relationship with the ripening and microbial action [31]; thus, they were related to raw material and ingredients supporting that no significant differences were detected between batches in most of the compounds. The levels of terpenes studied had a tendency to increase during the ripening process, probably due to dehydration: the loss of water causes the increase in fat level, in which terpenes are easily dissolved [21].

### 3.2. Texture of Ripened “Salchichón”

Results concerning the texture profile analysis of “salchichón” at the end of ripening showed similar values between *L. sakei* 205 inoculated and non-inoculated sausages (Table 3). Both batches showed lower levels of adhesiveness (−7.77 and −8.43) than other previous works (from −0.6 to −1.3 N s) [32,33]. The profile of texture analysis of both Control and Ls batches ranged in the usual values of dry-cured fermented sausages [32,34].

### 3.3. Color of Ripened “Salchichón”

Color determination is the first parameter that affects consumer acceptance of meat products. In the CIELAB parameters, no differences between the *L. sakei* 205 inoculated and Control batches were observed, except for the parameter L* (lightness) which was significantly (*p* ≤ 0.05) higher in the Ls than in Control batch (Table 3). A positive correlation between L* value and fat content, as well as a negative correlation with drying/ripening time, was previously reported [35]. However, a* (redness) and b* (yellowness) values of the Ls batch were very similar to those found in the Control batch. Thus, only differences in the color of sausages regarding the L* parameter could be attributed to inoculation with *L. sakei* 205. This agrees with previous studies that demonstrated that the application of LAB on a meat surface slightly increased its lightness (L*) [36]. These results are similar to those obtained by Kaban et al. [37] who did not find differences in a* and b* parameters when dry fermented sausages were inoculated with LAB and by Álvarez et al. [33] when *Enterococcus faecium* was inoculated into Iberian dry-cured fermented sausages.

### 3.4. Sensory Evaluation of the Ripened “Salchichón”

According to the levels of significance for the triangular test [38], more than 13 of the 24 panelists should detect differences between the Control and Ls batches to be considered as batches significantly different. However, only 9 out of 24 panelists observed differences between both batches. Thus, there were no significant differences (*p* > 0.05) between both Control and Ls batches. Thus, the selected strain of *L. sakei* 205 at the levels 6 log CFU/g inoculated has no significant effect on the sensory quality of the finished “salchichón”. No effect was also found by the use of selected LAB in sensorial analysis by triangular test in other meat products [39]. These results are consistent with the no differences observed in color and texture analysis in both Control and Ls batches. In addition, the increase in volatile compounds derived from amino acid catabolism and the decrease in compounds derived from lipid oxidation, observed throughout the ripening time, due to *L. sakei* 205 inoculation, was not enough to be differentiated from non-inoculated ones after sensorial analysis. In further studies, it could be investigated if, with a longer ripening process of “salchichón” inoculated with *L. sakei* 205, these differences in the volatile compounds could be enough to lead to positive effects in the sensorial analysis [40,41]. In any case, the results obtained on the non-modification of the sensory characteristics of the “salchichón” are relevant to propose *L. sakei* 205 as a protective culture.

## 4. Conclusions

The inoculation of selected *L. sakei* 205 does not cause a modification in the texture and color parameters. However, an increase in volatile compounds derived from amino acid catabolism and microbial esterification and a decrease in compounds derived from lipid oxidation were found throughout the ripening time, which could contribute positively to the flavor of “salchichón”. Thus, the use of *L. sakei* 205 as a protective culture could be advisable to increase the generation of volatile compounds associated with the characteristic flavor of traditional dry-cured fermented sausages.

## Figures and Tables

**Figure 1 biology-12-00088-f001:**
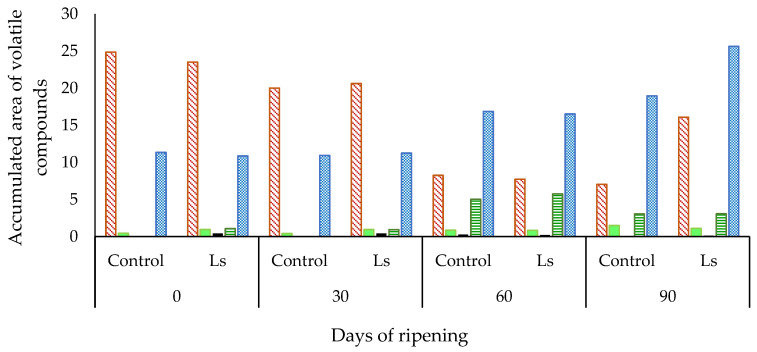
Accumulated area of volatile (×10^6^) compounds according to their origin, throughout the ripening of dry-cured fermented “salchichón” in Control and Ls (inoculated with *Lactilactobacillus sakei*) batches.

**Table 1 biology-12-00088-t001:** Profiles of volatile compounds (lipid oxidation and microbial esterification) of Control (uninoculated batch) and Ls (batch inoculated with *Lactilactobacillus sakei*) batches during the ripening process (0, 30, 60, and 90 days) of dry-cured fermented sausages (Arbitrary Area Units × 10^6^).

Origin/Compound	Control Batch				Ls Batch			
	0	30	60	90	0	30	60	90
**Lipid Oxidation**								
Tridecane	0.49 ± 0.10	0.35 ± 0.13	n.d.	n.d.	n.d.	n.d.	n.d.	n.d.
Dodecane	0.48 ± 0.07	0.34 ± 0.12	n.d.	n.d.	n.d.	n.d.	n.d.	0.11 ± 0.16
1-Pentanol	0.79 ± 0.20 ^a^	0.59 ± 0.24 ^a^	0.13 ± 0.01 ^b^	0.16 ± 0.15 ^b^	0.79 ± 0.04 ^a^	0.71 ± 0.26 ^ab^	0.13 ± 0.01 ^c^	0.34 ± 0.24 ^bc^
1-Hexanol	0.14 ± 0.13 ^b2^	0.08 ± 0.17 ^b2^	0.52 ± 0.19 ^ab^	1.06 ± 0.65 ^a^	2.94 ± 0.47 ^a1^	2.32 ± 1.16 ^a1^	0.41 ± 0.06 ^b^	1.66 ± 1.30 ^ab^
1-Heptanol	0.05 ± 0.11 ^2^	0.09 ± 0.12 ^2^	n.d.	0.07 ± 0.13	0.25 ± 0.04 ^a1^	0.07 ± 0.15 ^ab^	0.04 ± 0.09 ^ab^	0.20 ± 0.21 ^b^
1-Octanol	0.25 ± 0.05	0.11 ± 0.15	0.21 ± 0.07	0.08 ± 0.17	0.18 ± 0.10	0.16 ± 0.15	0.17 ± 0.06	0.39 ± 0.34
1-Octen-3-ol	1.60 ± 0.21 ^a^	1.20 ± 0.40 ^a^	0.57 ± 0.02 ^b1^	0.42 ± 0.34 ^b^	1.47 ± 0.11 ^a^	1.37 ± 0.26 ^a^	0.49 ± 0.07 ^b2^	0.76 ± 0.56 ^b^
2-Octen-1-ol, (E)-	0.33 ± 0.06 ^1^	0.27 ± 0.03	n.d.	n.d.	0.24 ± 0.02 ^2^	0.25 ± 0.03	n.d.	n.d.
Terpinen-4-ol	1.50 ± 0.02	1.33 ± 0.14	n.d.	n.d.	1.17 ± 0.04	n.d.	n.d.	n.d.
Pentanal	0.53 ± 0.12 ^a1^	0.38 ± 0.23 ^ab^	0.13 ± 0.01 ^b^	0.25 ± 0.01 ^b^	0.35 ± 0.07 ^2^	0.35 ± 0.07	0.18 ± 0.06	0.29 ± 0.28
Hexanal	12.36 ± 2.45 ^a1^	9.88 ± 3.05 ^a^	2.40 ± 0.10 ^b^	1.91 ± 2.24 ^b^	8.06 ± 1.40 ^a2^	8.00 ± 1.34 ^a^	1.94 ± 0.66 ^b^	5.14 ± 3.76 ^ab^
Heptanal	0.76 ± 0.10 ^a2^	0.66 ± 0.19 ^ab^	0.30 ± 0.10 ^bc^	0.40 ± 0.33 ^c^	0.93 ± 0.07 ^a1^	0.83 ± 0.20 ^a^	0.23 ± 0.06 ^b^	0.74 ± 0.58 ^ab^
2-Heptenal, (Z)-	0.39 ± 0.05 ^a^	0.36 ± 0.13 ^a^	0.21 ± 0.03 ^b^	n.d.	0.37 ± 0.04 ^b^	0.55 ± 0.09 ^a^	0.19 ± 0.05 ^c^	n.d.
2-Undecenal	0.31 ± 0.06 ^b^	0.26 ± 0.04 ^b^	0.46 ± 0.10 ^a2^	0.23 ± 0.02 ^b^	0.29 ± 0.02 ^b^	0.27 ± 0.04 ^b^	0.63 ± 0.07 ^a1^	0.93 ± 0.99 ^ab^
2-Nonenal, (E)-	0.80 ± 0.15	0.60 ± 0.24	n.d.	n.d.	0.86 ± 0.06	0.79 ± 0.21	n.d.	n.d.
Piperonal	0.31 ± 0.08 ^a^	0.25 ± 0.03 ^ab^	0.16 ± 0.03 ^b^	n.d.	0.31 ± 0.08	0.24 ± 0.03	0.23 ± 0.09	2.00 ± 1.78
2-Decenal,(Z)-	0.43 ± 0.04	0.36 ± 0.09	0.36 ± 0.11	n.d.	0.41 ± 0.02	0.31 ± 0.18	0.45 ± 0.08	0.30 ± 0.13
2-Octenal (E)-	1.01 ± 0.28 ^a^	0.71 ± 0.35 ^ab^	0.22 ± 0.04 ^b^	0.39 ± 0.02 ^b^	0.68 ± 0.06 ^a^	0.62 ± 0.16 ^a^	0.24 ± 0.06 ^b^	0.52 ± 0.60 ^ab^
2,4- Decadienal, (E,E)-	0.24 ± 0.02	0.21 ± 0.02	0.32 ± 0.22	n.d.	0.23 ± 0.02	0.23 ± 0.02	0.35 ± 0.06	0.22 ± 0.02
2-Heptanone	n.d.	n.d.	0.56 ± 0.03	0.55 ± 0.06	0.87 ± 0.14 ^a^	0.53 ± 0.36 ^b^	0.48 ± 0.06 ^b^	0.52 ± 0.01 ^b^
2-Octanone	n.d.	n.d.	0.17 ± 0.03 ^b^	0.32 ± 0.10 ^a^	0.30 ± 0.07 ^a^	0.25 ± 0.02 ^a^	0.14 ± 0.01 ^b^	0.28 ± 0.01 ^a^
2-Nonanone	n.d.	n.d.	0.30 ± 0.07	0.28 ± 0.01	0.31 ± 0.08 ^a^	0.19 ± 0.11 ^b^	0.20 ± 0.12 ^b^	0.28 ± 0.01 ^b^
Hexanoic acid	1.89 ± 0.56 ^a^	1.73 ± 0.71 ^a^	0.59 ± 0.10 ^b^	0.68 ± 0.17 ^b^	1.96 ± 0.21 ^a^	1.96 ± 0.55 ^a^	0.60 ± 0.11 ^b^	1.52 ± 0.85 ^ab^
Octanoic acid	n.d.	n.d.	0.50 ± 0.49	0.31 ± 0.17 ^a^	n.d.	n.d.	0.50 ± 0.46	0.19 ± 0.26 ^a^
Butanoic acid	0.19 ± 0.11 ^b2^	0.28 ± 0.02 ^a2^	0.13 ± 0.01 ^b^	0.13 ± 0.12 ^b^	0.57 ± 0.04 ^a1^	0.64 ± 0.08 ^a1^	0.13 ± 0.01 ^b^	0.23 ± 0.15 ^b^
**Microbial esterification**								
Hexanoic acid, ethyl ester	n.d.	n.d.	n.d.	0.34 ± 0.40 ^1^	0.44 ± 0.11 ^a^	0.41 ± 0.06 ^a^	n.d.	0.26 ± 0.05 ^b2^
n-Caproic acid vinyl ester	0.45 ± 0.09 ^a1^	0.41 ± 0.09 ^ab^	0.17 ± 0.02 ^c^	0.24 ± 0.01 ^bc^	0.31 ± 0.02 ^a2^	0.34 ± 0.06 ^a^	0.19 ± 0.07 ^b^	0.27 ± 0.03 ^ab^
Octanoic acid, ethyl ester	n.d.	n.d.	0.69 ± 0.14	0.91 ± 0.27 ^1^	0.21 ± 0.01 ^b^	0.21 ± 0.01 ^b^	0.66 ± 0.28 ^a^	0.56 ± 0.12 ^ab2^

Control (uninoculated batch), Ls (batch inoculated with *L. sakei*). Values of volatile compounds are expressed as mean ± standard deviation of the analysis made in quintuplicate. The means with different letters (a–c) in the same row indicate significant differences (*p* ≤ 0.05) between days in the same batch and the same compound. Mean values with different numbers (1–2) in the same column indicate significant differences (*p* ≤ 0.05) between batches on the same day. n.d. means not detected.

**Table 2 biology-12-00088-t002:** Profiles of volatile compounds (carbohydrate fermentation, amino acid catabolism, spices, and others) of Control (uninoculated batch) and Ls (batch inoculated with *Lactilactobacillus sakei*) during the ripening process (0, 30, 60, and 90 days) of dry-cured fermented sausages (Arbitrary Area Units × 10^6^).

Origin/Compound	Control				Inoculated Ls 205			
	0	30	60	90	0	30	60	90
**Carbohydrate Fermentation**								
Acetic acid	0.04 ± 0.09 ^b2^	n.d.	0.27 ± 0.03 ^a^	n.d.	0.42 ± 0.04 ^a1^	0.43 ± 0.10 ^a^	0.22 ± 0.04 ^b^	0.12 ± 0.17 ^b^
**Amino Acid Catabolism**								
2-methyl-1-propanol	n.d.	n.d.	1.99 ± 0.15 ^a^	0.98 ± 0.11 ^b1^	0.27 ± 0.04 ^b^	0.09 ± 0.13 ^b^	1.99 ± 0.44 ^a^	0.55 ± 0.17 ^b2^
3-methyl-butanal	n.d.	n.d.	1.40 ± 0.72 ^a^	0.10 ± 0.21 ^b^	0.83 ± 0.16 ^b^	n.d.	0.88 ± 0.85 ^a^	n.d.
3-methyl-1-butanol	n.d.	n.d.	1.03 ± 0.10	1.35 ± 0.57	n.d.	0.84 ± 0.15 ^b^	1.25 ± 0.20 ^a^	1.18 ± 0.28 ^a^
2-methyl-propanoic acid	n.d.	n.d.	0.13 ± 0.01	0.10 ± 0.15	n.d.	n.d.	0.52 ± 0.30	0.51 ± 0.38
3-methyl-butanoic acid	n.d.	n.d.	0.18 ± 0.05	0.28 ± 0.39	n.d.	n.d.	0.71 ± 0.63	0.74 ± 0.45
2-methyl-1-butanol	n.d.	n.d.	0.31 ± 0.05 ^a2^	0.24 ± 0.02 ^b^	n.d.	n.d.	0.40 ± 0.03 ^a1^	0.09 ± 0.13 ^b^
**Spices**								
α-pinene	0.22 ± 0.21	0.17 ± 0.20	0.53 ± 0.12	0.56 ± 0.09	0.18 ± 0.21	0.27 ± 0.196	0.24 ± 0.33	0.25 ± 0.10
β-pinene	1.04 ± 0.09 ^b^	1.01 ± 0.25 ^b^	1.87 ± 0.26 ^a^	2.12 ± 0.65 ^a^	1.07 ± 0.10 ^b^	1.10 ± 0.28 ^b^	2.18 ± 0.46 ^a^	1.75 ± 0.20 ^a^
α-terpineol	0.34 ± 0.02 ^a1^	0.28 ± 0.02 ^b^	0.25 ± 0.08 ^b^	0.28 ± 0.04 ^b^	0.25 ± 0.03 ^2^	0.20 ± 0.11	0.26 ± 0.03	0.31 ± 0.07
Safrole	0.58 ± 0.07 ^1^	0.55 ± 0.08	0.58 ± 0.08	0.57 ± 0.03	0.48 ± 0.04 ^b2^	0.47 ± 0.05 ^b^	0.56 ± 0.02 ^a^	0.51 ± 0.03 ^ab^
D-limonene	2.28 ± 0.13 ^b^	2.19 ± 0.43 ^b^	4.45 ± 0.97 ^b^	5.94 ± 2.37 ^a^	2.20 ± 0.30 ^b^	2.30 ± 0.72 ^b^	4.44 ± 1.12 ^b^	12.67 ± 8.63 ^a^
o-Cymene	0.55 ± 0.03 ^2^	0.45 ± 0.05 ^2^	0.76 ± 0.21	0.76 ± 0.68	1.39 ± 0.34 ^ab1^	1.33 ± 0.39 ^ab1^	0.60 ± 0.17 ^b^	3.37 ± 2.75 ^a^
3-carene	1.85 ± 0.10 ^b^	1.72 ± 0.33 ^b^	2.65 ± 0.34 ^a^	2.73 ± 0.10 ^a^	1.66 ± 0.32 ^b^	1.65 ± 0.62 ^b^	2.68 ± 0.24 ^a^	1.86 ± 1.13 ^ab^
(+)-4-Carene	n.d.	0.24 ± 0.01 ^b^	0.19 ± 0.03 ^c^	0.32 ± 0.01 ^a^	n.d.	0.24 ± 0.01 ^b^	0.20 ± 0.01 ^c^	0.32 ± 0.01 ^a^
α-phellandrene	n.d.	0.04 ± 0.10 ^b^	0.40 ± 0.09 ^a^	0.39 ± 0.11 ^a^	n.d.	0.05 ± 0.11 ^b^	0.48 ± 0.08 ^a^	0.43 ± 0.27 ^a^
β-phellandrene	0.33 ± 0.21 ^ab^	0.21 ± 0.14 ^b^	0.53 ± 0.16 ^a^	0.63 ± 0.39 ^a^	0.26 ± 0.03 ^bc^	0.25 ± 0.06 ^c^	0.55 ± 0.02 ^ab^	0.55 ± 0.19 ^a^
α-copaene	0.31 ± 0.03 ^1^	0.28 ± 0.04	0.27 ± 0.04	0.27 ± 0.02	0.24 ± 0.03 ^2^	0.24 ± 0.03	0.26 ± 0.01	0.24 ± 0.02
Caryophyllene	3.58 ± 0.48 ^1^	3.56 ± 0.65	4.12 ± 0.57	4.15 ± 0.31 ^1^	2.92 ± 0.34 ^b2^	2.96 ± 0.35 ^b^	4.07 ± 0.13 ^a^	3.15 ± 0.53 ^b2^
Humulene	0.24 ± 0.02	0.25 ± 0.05	0.26 ± 0.04	0.23 ± 0.02 ^1^	0.20 ± 0.01	0.20 ± 0.01	n.d.	0.21 ± 0.01 ^2^
**Others**								
Caryophyllene oxide	0.69 ± 0.03 ^a1^	0.40 ± 0.04 ^b^	n.d.	n.d.	0.41 ± 0.07 ^2^	0.39 ± 0.10	n.d.	0.41 ± 0.13
Cyclohexene, 4-ethenyl-4-methyl-3-(1-methylethenyl)-1-(1-methylethyl)-, (3R-trans)-	0.29 ± 0.03 ^b^	0.25 ± 0.03 ^b^	0.54 ± 0.04 ^a^	0.29 ± 0.02 ^b^	0.22 ± 0.02 ^b^	0.25 ± 0.01 ^b^	0.49 ± 0.05 ^a^	0.24 ± 0.01 ^b^
1,3-Benzodioxole, 4-methoxy-6-(2-propenyl)-	0.97 ± 0.12 ^b^	0.93 ± 0.11 ^b^	1.08 ± 0.45 ^a^	0.96 ± 0.08 ^b^	0.84 ± 0.04 ^b^	0.80 ± 0.08 ^b^	0.93 ± 0.05 ^a^	n.d.

Control (uninoculated batch), Ls (batch inoculated with *L. sakei*). Values are expressed as mean ± standard deviation of the analysis made in quintuplicate. The means with different letters (a–c) in the same row indicate significant differences (*p* ≤ 0.05) between days in the same batch and the same compound. Mean values with different numbers (1–2) in the same column indicate significant differences (*p* ≤ 0.05) between batches on the same day. n.d. means not detected.

**Table 3 biology-12-00088-t003:** Values of instrumental texture parameters (hardness, adhesiveness, springiness, cohesiveness, and chewiness) and CIE L* (lightness) a* (redness) b* (yellowness) (CIELAB) of dry-cured fermented sausages (“salchichón”) at the end of the ripening process.

Parameters	Batches	
	Control	Ls
Hardness (N)	214.78 ± 45.43	220.32 ± 29.13
Adhesiveness (N/s)	−7.77 ± 1.41	−8.43 ± 2.05
Springiness	0.67 ± 0.09	0.63 ± 0.07
Cohesiveness	0.61 ± 0.02	0.60 ± 0.01
Chewiness (N)	89.44 ± 26.75	82.82 ± 16.57
L*	36.62 ± 0.62	37.83 ± 0.79 *
a*	12.38 ± 1.14	11.63 ± 0.28
b*	4.23 ± 0.49	4.55 ± 0.56

Control (uninoculated batch), Ls (batch inoculated with *Lactilactobacillus sakei*). Values are expressed as mean ± standard deviation. Asterisks indicate significant differences with respect to the Control (*p* ≤ 0.05).

## Data Availability

Not applicable.

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
