# Peer review of "Effect of a Selected Protective Culture of *Lactilactobacillus sakei* on the Evolution of Volatile Compounds and on the Final Sensorial Characteristics of Traditional Dry-Cured Fermented “Salchichón”"

_biology, 2023, doi:10.3390/biology12010088_

Round 1

Reviewer 1 Report

Summary

In this article the authors assess a strain of L. sakei, used as a ‘protective culture’ in the production of “salchichón”. 

General Comments

While the manuscript covers the background on “salchichón” and the reasoning behind the potential need for a ‘protective culture’, the data provided (along with the interpretation) doesn't particularly add anything new to the field while the lack of analysis of strains present during the fermentation period is a major flaw which cannot be overlooked when the purpose of the work is to suggest the use of a ‘protective culture’. Possibly, with new experiments and major revisions (including identification and quantification of strains present throughout the fermentation and the levels of Listeria monocytogenes) it could be considered to be submitted to another mdpi journal, such as Foods or Fermentation.

Author Response

In our opinion, the work shows new information about the evolution of volatile compounds in dry-cured fermented sausages ripened with selected protective L. sakei throughout the ripening and also the influence of this selected strain in the final characteristic of ripened “salchichón”. The microbial population present in the dry-cured fermented sausages “salchichón” throughout the ripening process has been studied and it has been demonstrated that L. sakei 205 is able to survive and colonize the product. From the results, it is demonstrated that the addition of a protective culture of selected L. sakei to the “salchichón” does not have a negative impact on the sensorial characteristic of the product, but it can even have a positive effect on the formation of volatile compounds related to characteristic flavor development of “salchichón”.

Reviewer 2 Report

The manuscript by Martin et al. is well designed and conducted, having focused on the beneficial effects of a selected culture of lactic acid bacteria (L. sakei 205) on the evolution of volatile compounds and on the sensory characteristics of a traditional dry fermented product, namely "salchichón". I support its publication after appropriate minor modifications as outlined below.

Introduction:

In order to increase the reader's interest, I suggest including a brief description of the 'salchichón' product (sensory characteristics, physico-chemical composition) and a diagram of the processing technology.

Please insert the adequate references for "U.S. Food and Drug Administration (FDA) and the Qualified Presumption of Safety (QPS) from the European Food Safety Authority (EFSA) " (lines 53-56).

Materials and methods:

- subchapter 2.1. the authors should replace “according to Martín et al., [4]” with “according to the method described by Martin et al. [4]” (line 76);

- subchapter 2.2. No any reference was inserted or the used methods were original? Please insert the adequate references;

- subchapter 2.3.; 2.4; 2.5; 2.6. No any reference work was inserted. Please provide references for the testing protocols.

Please extend the Discussion chapter, including the comparation of the study results with the results from other significant references.

In the Conclusion chapter, the authors state that “the use of L. sakei 205 as a protective culture could be advisable not only by to assure the innocuity of the product... ”. However, this research didn’t aim to evaluate the potential of L. sakei 205 to ensure the innocuity of the 'salchichón' product. Please rewrite this section focusing only on a presentation of conclusions derived from the obtained results.

Author Response

The manuscript by Martin et al. is well designed and conducted, having focused on the beneficial effects of a selected culture of lactic acid bacteria (L. sakei 205) on the evolution of volatile compounds and on the sensory characteristics of a traditional dry fermented product, namely "salchichón". I support its publication after appropriate minor modifications as outlined below.

Response: We thank Reviewer 2 for the positive comments on the manuscript.

Introduction:

In order to increase the reader's interest, I suggest including a brief description of the 'salchichón' product (sensory characteristics, physico-chemical composition) and a diagram of the processing technology.

Response: As suggested by Reviewer 2, a brief description of the “salchichón” product has been included in the Introduction section of the revised manuscript. We consider that a diagram of the processing technology may confuse readers since this process is quite artisanal and can be different between manufacturers. For this reason, we did not include the diagram in the revised version.

Please insert the adequate references for "U.S. Food and Drug Administration (FDA) and the Qualified Presumption of Safety (QPS) from the European Food Safety Authority (EFSA) " (lines 53-56).

Response: The adequate reference has been included.

Materials and methods:

- subchapter 2.1. the authors should replace “according to Martín et al., [4]” with “according to the method described by Martin et al. [4]” (line 76);

Response: The sentence “according to Martín et al., [4]” has been replaced by “according to the method described by Martin et al. [4]”.

- subchapter 2.2. No any reference was inserted or the used methods were original? Please insert the adequate references;

Response: The used methods are optimised and developed by our research group. The reference Martín et al. (2021) (Effect of the dry-cured fermented sausage “salchichón” processing with a selected Lactobacillus sakei in Listeria monocytogenes and microbial population. Foods 2021, 10, doi:10.3390/foods10040856) has been included in the Material and Methods section of the revised manuscript.

- subchapter 2.3.; 2.4; 2.5; 2.6. No any reference work was inserted. Please provide references for the testing protocols.

Response:  The required references have been inserted in subchapter 2.3, 2.4, 2.5 and 2.6 of the revised manuscript.

Pons, M.; Fiszman, S.M. Instrumental texture profile analysis with particular reference to gelled systems. J. Texture Stud. 1996, 27, 597–624, doi:10.1111/j.1745-4603.1996.tb00996.x.

Clydesdale, F.M.; Ahmed, E.M. Colorimetry — methodology and applications. Food Sci. Nutr. 1978, 10, 37–41.

Please extend the Discussion chapter, including the comparation of the study results with the results from other significant references.

Response: The Discussion chapter has been extended and we have included the comparison of our results with those of other authors.

In the Conclusion chapter, the authors state that “the use of L. sakei 205 as a protective culture could be advisable not only by to assure the innocuity of the product... ”. However, this research didn’t aim to evaluate the potential of L. sakei 205 to ensure the innocuity of the 'salchichón' product. Please rewrite this section focusing only on a presentation of conclusions derived from the obtained results.

Response: The Conclusion section has been modified to focus only on the presentation of the assessments derived from the obtained results.

Reviewer 3 Report

The present study examined the effect of Lactilactobacillus sakei on the production of volatile compounds, as well as the sensory qualities of traditional dry-cured fermented “salchichón”

Major comments

1.         The manuscripts have flaws in language. It needs seriously language editing. Errors in sentence structure should be corrected. I suggest that the authors use the services of a native English speaker.

2.         Introduction, discussion and conclusion section need to be improved.  The authors failed to highlight the impact of the present study considering that they observed no significant difference in the parameters examined for both the inoculated batches and the control batches.

Comments

Line 15, “…is not extended due to….” Extended to what? What are the negative effects? What quality? Safety or sensory qualities? The authors should write in clear manner.

Line 17-18, correct sentence

Line 22, “… con…”?

Line 25-26, Correct sentence “The inoculation of…”

Line 31, Define LAB – first time use in the paper

Line 42-43, correct sentence “This problem could…”

Line 44-53, correct error in the sentence.

Line 60, change “…of this product.” To “…of the final product”.

Line 60, change “…taking in account…” to “…taking into account…”

Line 68-70, correct sentence

Line 72, 100 µL of what?

Line 73, the comma should come before the “and”

Line 78-80, correct error in sentence

Line 86, define the abbreviation “PBS”

Line 92, change “…processing in…” to “…processing conditions in…”

Line 100, delete “at”

Line 103-104, correct the sentence

Line 149-151, correct the error in sentence. Which variables are dependent and independent?

Line 152 – 154, correct sentence.

Line 160, “…higher levels of adhesiveness…”. State the values

Line 162, ad? Correct accordingly

Line 205-226, correct errors in the paragraph

Line 292, suppose?

Table 1, what is CIELAB?  write CIELAB in full. Define L*, a*, and b* in the footnote

Table 2, write Ls in the title in full. What are the numbers in the 2nd row, number of days? Define n.d. in the footnote.

Author Response

The present study examined the effect of Lactilactobacillus sakei on the production of volatile compounds, as well as the sensory qualities of traditional dry-cured fermented “salchichón”

Major comments

  1. The manuscripts have flaws in language. It needs seriously language editing. Errors in sentence structure should be corrected. I suggest that the authors use the services of a native English speaker.

Response: As suggested by Reviewer 3, the English writing of the manuscript has been revised and corrected by a native speaker.

  1. Introduction, discussion and conclusion section need to be improved. The authors failed to highlight the impact of the present study considering that they observed no significant difference in the parameters examined for both the inoculated batches and the control batches.

Response: As suggested by Reviewer 3, the introduction, discussion, and conclusion sections have been improved in the revised version of the manuscript. In addition, the impact of study has been highlighted.

Comments

Line 15, “…is not extended due to….” Extended to what? What are the negative effects? What quality? Safety or sensory qualities? The authors should write in clear manner.

Response: The word extended has been changed by widespread for easier understanding. “Sensory quality” has been indicated to clarify the meaning of the sentence.

Line 17-18, correct sentence

Response: This sentence has been corrected.

Line 22, “… con…”?

Response: This error has been corrected in the new version of the manuscript

Line 25-26, Correct sentence “The inoculation of…”

Response: The sentence has been corrected.

Line 31, Define LAB – first time use in the paper

Response: LAB has been defined the first time used in the paper.

Line 42-43, correct sentence “This problem could…”

Response: The sentence has been corrected.

Line 44-53, correct error in the sentence.

Response: This sentence has been modified in the revised version.

Line 60, change “…of this product.” To “…of the final product”.

Line 60, change “…taking in account…” to “…taking into account…”

Response: As Reviewer 3 requested, the above changes have been made.

Line 68-70, correct sentence

Response: The sentence has been corrected.

Line 72, 100 µL of what?

Response: 100 µL of the inoculum. This has been included in the new version of the manuscript.

Line 73, the comma should come before the “and”

Response: The comma has been inserted before the “and”.

Line 78-80, correct error in sentence

Response: This sentence has been corrected in the new version of the manuscript.

Line 86, define the abbreviation “PBS”

Response: The abbreviation “PBS” has been defined in the new version of the manuscript.

Line 92, change “…processing in…” to “…processing conditions in…”

Response: This change has been included in the revised version of the manuscript.

Line 100, delete “at”

Response: The word “at” has been deleted in the new version of the manuscript.

Line 103-104, correct the sentence

Response: This sentence has been corrected in the revised version of the manuscript.

Line 149-151, correct the error in sentence. Which variables are dependent and independent?

Response: The sentence has been corrected and the dependent and independent variables have been identified.

Line 152 – 154, correct sentence.

Response: The sentence has been corrected in the new version on the manuscript.

Line 160, “…higher levels of adhesiveness…”. State the values

Response: The Reviewer 3 is right for this reason the sentence has been changed to “lower levels of adhesiveness”. In addition, these levels have been stated.

Line 162, ad? Correct accordingly

Response: The error has been corrected.

Line 205-226, correct errors in the paragraph

Response: The sentence has been corrected.

Line 292, suppose?

Response: The verb has been modified by “cause” in the new version of the manuscript

Table 1, what is CIELAB? Write CIELAB in full. Define L*, a*, and b* in the footnote

Response: CIELAB has been fully written and L* a* and b* parameters have been defined.

Table 2, write Ls in the title in full. What are the numbers in the 2nd row, number of days? Define n.d. in the footnote.

Response: Ls has been written in the title in full. Yes, the number in the 2nd row are the number of days. To avoid misunderstanding this has been specified in the title of the Table 2. Finally, n.d. has been defined in the footnote.

Reviewer 4 Report

The manuscript entitled 'Effect of a selected protective culture of Lactilactobacillus sakei on the evolution of volatile compounds and on the final senso-3 rial characteristics of traditional dry-cured fermented “sal-4 chichón' explored the effect of Lactilactobacillus sakei on the production of traditional “salchichón”. Overall, the content of the manuscript is clear and well written. Considering as above, the present version can be improved as the following points:

1. In line 77, the adding amount of strain was out of references, L. sakei at 7 log CFU/g (Ls) was adopted from which method. Please have a check.

2. In Table 1, the Cohesiveness of LS was 0.6, and control was 0.61, how did you analyze the data to have this differences between Ls and control, please have a check again.

3. In the color results,the reason of increasing L* values in the Ls groups was out of discussion.

4.  How many repetitions in Table 2, please marked it in the end of table.

5.  How many repetitions in Figure 1, where the filling form of the bar chart was not very clear. In addition, the 0-90 on the horizontal axis needs to be labeled as days.

6. In line 277-289, the sensory quality was measured according to the levels of significance for the triangular test, but where is the data in your discussion.

Author Response

The manuscript entitled 'Effect of a selected protective culture of Lactilactobacillus sakei on the evolution of volatile compounds and on the final sensorial characteristics of traditional dry-cured fermented “salchichón' explored the effect of Lactilactobacillus sakei on the production of traditional “salchichón”. Overall, the content of the manuscript is clear and well-written. Considering as above, the present version can be improved on the following points:

  1. In line 77, the adding amount of strain was out of references, L. sakei at 7 log CFU/g (Ls) was adopted from which method. Please have a check.

Response: Reviewer 4 is right; we added L. sakei at 6 log CFU/g. This has been corrected in the revised version of the manuscript.

  1. In Table 1, the Cohesiveness of LS was 0.6, and control was 0.61, how did you analyze the data to have these differences between Ls and control, please have a check again.

Response: As Reviewer 4 suggested, we checked the values of cohesiveness between both batches and no differences were found. This has been amended in the revised manuscript.

  1. In the color results, the reason of increasing L* values in the Ls groups was out of discussion.

Response: The discussion about the increase of L* values in Ls batch has been included in the revised version of the manuscript.

  1. How many repetitions in Table 2, please marked it in the end of table.

Response: This study was performed in quintuplicate. This has been included in the Material and Methods section and also in Tables 2 and 3 of the revised manuscript.

  1. How many repetitions in Figure 1, where the filling form of the bar chart was not very clear. In addition, the 0-90 on the horizontal axis needs to be labeled as days.

Response: Figure 1 is a summation of the quintuplicate of the analysis of volatile compounds. As suggested by reviewer 4, the authors have introduced the outline of the bars to make them clearer.

  1. In line 277-289, the sensory quality was measured according to the levels of significance for the triangular test, but where is the data in your discussion.

Response: According to Reviewer 4, the data of the significance for the triangular test has been included in Results and Discussion section of the revised manuscript.

Reviewer 5 Report

Overall comments:

The manuscript evaluated the effect of a selected protective culture of Lactilactobacillus sakei 205 on the evolution of volatile compounds throughout the ripening process and on the final sensorial characteristics of traditional dry-cured fermented “salchichón”. Although the “salchichón” is a niche food worldwide, it can still be regarded as a typical study of sausage fermentation. The extensive volatile compounds involved in the ripening process of salchichón were analyzed through GC-MS, which is a highlight of the manuscript and may be of interest to the reader. However, there are still several issues need to be addressed.

1, The reviewer agreed that the flavor of the fermented product including sausage varies from place to place. As the author said in the Introduction section, the addition of safe protective cultures (LAB) provides a way to guarantee the special flavor of the final product. Even if sausages are produced in other regions, the inoculation of specific LAB can ensure that the typical flavor does not change. While, this research topic was not studied in the manuscript. Thus, the content of Introduction section should be adjust to fit the Results and Discussion section.

2, The English writing of the manuscript should be further improved by native speaker or language editing serves.

3, Only two Table are provided in the manuscript, it is not enough.

4, "The inoculation of selected L. sakei 205 contributed positively to flavor of salchichón" concluded by the author is not matched with the Sensory evaluation results. Please explain.

5, Double check the format of the manuscript, especially the format of Reference and Table sections.

Special comments:

L 40, the word 'contaminant' should be used with caution. The natural microbes are beneficial for the fermentation of sausage. Please modify it throughout the manuscript.

L 52, please provide the detailed information of in vivo assays. What are the research subjects?

L 52-53, please rewrite the sentence. The reviewer can not get the sense of '...being the last...'.

L 53-56, provide the related references.

L 74-75, please provide the proportion of dilution.

L 84, the full name of 'CFU' should be given at the first appearance.

L 86, the same as above.

L 101, how to reach a microaerophilic condition, please provide the equipment information if available.

L 103-104, please provide the Table showing the number of LAB.

L 111-112, please explain why the sensory analysis uses three repetitions instead of five.

L 116, '...2.5 cm. that...' ???

L 156, since the texture of ripened “salchichón” was analyze at the end of fermentation, it should be placed after the subtitle of 'Volatile compound analysis' in Materials and Methods and Results and Discussion sections. 

L 158-160, please explain what a lower cohesiveness value and a higher adhesiveness value means, is it better or worse?

L 164, the full name of 'CIELAB' should be given at the first appearance.

L 174-175, what is negatively correlated with the drying/ripening time? Please write it clearly.

L 176-178, is there a causal relationship between the two parts of the sentence? Please explain.

L 192, Are amino acid catabolism increasing throughout the ripening time? Please check.

L 180-195, please provide the comparison and discussion between batches.

Split Table 2 because it is too large.

L 208, why bold fonts?

L 217-218, The reviewer do not agree that the formation of hexal is only associated with lactic acid bacteria. Other microbes, apparently, are also at work during the hexal formation and sausage fermentation. Please explain.

L 220-221, whether the typical flavor was good or not? If it was accepted or even liked, whether hexanal should not be decreased. Please explian.

L 222-223, the discussion here is repeated as above part.

L 243-244, how to conclude 'the same level', please explain.

L 272, please provide the related references.

Author Response

Overall comments:

The manuscript evaluated the effect of a selected protective culture of Lactilactobacillus sakei 205 on the evolution of volatile compounds throughout the ripening process and on the final sensorial characteristics of traditional dry-cured fermented “salchichón”. Although the “salchichón” is a niche food worldwide, it can still be regarded as a typical study of sausage fermentation. The extensive volatile compounds involved in the ripening process of salchichón were analyzed through GC-MS, which is a highlight of the manuscript and may be of interest to the reader. However, there are still several issues need to be addressed.

Response: We thank Reviewer 5 for the positive comments on the manuscript.

1, The reviewer agreed that the flavor of the fermented product including sausage varies from place to place. As the author said in the Introduction section, the addition of safe protective cultures (LAB) provides a way to guarantee the special flavor of the final product. Even if sausages are produced in other regions, the inoculation of specific LAB can ensure that the typical flavor does not change. While, this research topic was not studied in the manuscript. Thus, the content of Introduction section should be adjust to fit the Results and Discussion section.

Response: The Introduction section has been rewritten to fit the Results and Discussion section.

2, The English writing of the manuscript should be further improved by native speaker or language editing serves.

Response: As suggested by Reviewer 5, the English writing of the manuscript has been revised by a native speaker.

3, Only two Table are provided in the manuscript, it is not enough.

Response: In the present work two Tables and one Figure are provided and, in our opinion, could be enough to show the results of this paper. In addition, the two Tables contained an important amount of data on volatile compounds of “salchichón” throughout the ripening process, which highlighted the results of this work.

4, "The inoculation of selected L. sakei 205 contributed positively to flavor of salchichón" concluded by the author is not matched with the Sensory evaluation results. Please explain.

Response: From the results the inoculation of L. sakei 205 contributed positively to the flavor development of “salchichón”, provoking differences in volatile compounds derived from amino acid catabolism and compounds derived from lipid oxidation, in comparison with Control batch. However, these differences were not sufficient in this work to be detected by the panellists in a triangular test. In further studies, it could be investigated if, with a longer ripening process of “salchichón” inoculated with L. sakei 205, these differences in the volatile compounds could be enough to lead to positive effects in the sensorial analysis. This has been added to the revised version of the manuscript.

5, Double check the format of the manuscript, especially the format of Reference and Table sections.

Response: The format of the manuscript has been checked.

Special comments:

L 40, the word 'contaminant' should be used with caution. The natural microbes are beneficial for the fermentation of sausage. Please modify it throughout the manuscript.

Response: Reviewer 5 is right, so the word "contaminant” has been removed in the corrected version of the manuscript.

L 52, please provide the detailed information of in vivo assays. What are the research subjects?

Response: As suggested by Reviewer 5, detailed information of the in vivo assays has been provided in the new version of the manuscript.

L 52-53, please rewrite the sentence. The reviewer can not get the sense of '...being the last...'.

Response: As suggested by Reviewer 5, the sentence has been rewritten.

L 53-56, provide the related references.

Response: The related references have been provided.

L 74-75, please provide the proportion of dilution.

Response: This has been provided in the revised version.

L 84, the full name of 'CFU' should be given at the first appearance.

Response: The full name of “CFU” has been given at the first appearance.

L 86, the same as above.

Response: As suggested by Reviewer 5, the full name of “PBS” has been included.

L 101, how to reach a microaerophilic condition, please provide the equipment information if available.

Response: Microaerophilic conditions were reached by using a double layer of MRS agar.

L 103-104, please provide the Table showing the number of LAB.

Response: In our opinion, the data given about the number of LAB in control and inoculated batches is enough, since the most important for the reader is that LAB counts were found at higher levels throughout the ripening process in the inoculated batch.

L 111-112, please explain why the sensory analysis uses three repetitions instead of five.

Response: The sensory analysis was carried out by 24 panelists, but since it was a triangular sensory analysis, 3 samples were presented to each panelist.

L 116, '...2.5 cm. that...' ???

Response: The point has been deleted

L 156, since the texture of ripened “salchichón” was analyze at the end of fermentation, it should be placed after the subtitle of 'Volatile compound analysis' in Materials and Methods and Results and Discussion sections.

Response: In our opinion, the Results and Discussion section was organized for a better understanding for the readers by showing first the results of the instrumental analysis (texture and color) and then volatile compounds and finally sensorial analysis

L 158-160, please explain what a lower cohesiveness value and a higher adhesiveness value means, is it better or worse?

Response: The cohesiveness is related to the consistency of the “salchichón”. A lower value in L. sakei inoculated batch than in control batch could be better for the chewiness of the inoculated product. However, no differences were found in the triangular test by the panelists. There were no significant differences between batches in adhesiveness.

L 164, the full name of 'CIELAB' should be given at the first appearance.

Response: The full name of CIELAB has been given in the new version of this manuscript.

L 174-175, what is negatively correlated with the drying/ripening time? Please write it clearly.

Response: The sentence has been rewritten to improve its understanding.

L 176-178, is there a causal relationship between the two parts of the sentence? Please explain.

Response: This sentence was removed from the revised version of the manuscript.

L 192, Are amino acid catabolism increasing throughout the ripening time? Please check.

Response: The accumulated area of compounds derived from amino acid catabolism in-creased throughout the ripening time in Ls batch showing the highest mean values at day 60. This has been checked and corrected in the revised version of the manuscript.

L 180-195, please provide the comparison and discussion between batches.

Split Table 2 because it is too large.

Response: As suggested by Reviewer 5, the comparison and discussion between batches have been provided and Table 2 has been split and divided into Tables 2 and 3.

L 208, why bold fonts?

Response: We have used bold font to make the volatile compound results easier to read.

L 217-218, The reviewer do not agree that the formation of hexal is only associated with lactic acid bacteria. Other microbes, apparently, are also at work during the hexal formation and sausage fermentation. Please explain.

Response: We agree with Reviewer 5 about the formation of hexanal. Thus, this paragraph has been rewritten and the reference to the formation of hexanal only by lactic-acid bacteria has been modified by microbial activity in the revised version of the manuscript.

L 220-221, whether the typical flavor was good or not? If it was accepted or even liked, whether hexanal should not be decreased. Please explain.

Response: The excessive formation of hexanal gives the dry-cured fermented meat products a flavor of rancid, pungent, and toasty. Thus, the reduction observed for this compound in both batches throughout the ripening process it is very interesting for the flavor development of the final product since may contribute to reducing the note of rancidity of ripened “salchichón”. This paragraph has been rewritten as above in the revised version of the manuscript.

L 222-223, the discussion here is repeated as above part.

Response: This repetition has been eliminated from the revised version.

L 243-244, how to conclude 'the same level', please explain.

Response: To avoid misunderstandings by the readers this sentence has been rewritten and “the same level” has been eliminated.

L 272, please provide the related references.

Response: The related reference (Bis- Souza et al., 2019) has been provided.

Bis-Souza, C.V.; Pateiro, M.; Domínguez, R.; Lorenzo, J.M.; Penna, A.L.B.; da Silva Barretto, A.C. Volatile profile of fermented sausages with commercial probiotic strains and fructooligosaccharides. J. Food Sci. Technol. 2019, 56, 5465–5473, doi:10.1007/s13197-019-04018-8.

Round 2

Reviewer 1 Report

The reviewer appreciates that considerable efforts have been made to improve the introduction and background in this manuscript. Significant changes have also been made to the methodology and results interpretation. However, the reviewer still believes that quantification of strains present throughout the fermentation (what are the levels of L. sakei in the control? How dominant is L. sakei in the inoculated samples over time?) and the levels of Listeria monocytogenes (stated as ‘the most worrying microorganism’) are required in order to complete the story of this manuscript. Also, despite some improvements to the language, further corrections are still required. The reviewer would still recommend the manuscript be submitted to another, most scientifically appropriate mdpi journal, such as Foods or Fermentation.

Author Response

The reviewer appreciates that considerable efforts have been made to improve the introduction and background in this manuscript. Significant changes have also been made to the methodology and results interpretation.

Response: We thank Reviewer 1 for the positive comments on the manuscript.

However, the reviewer still believes that quantification of strains present throughout the fermentation (what are the levels of L. sakei in the control? How dominant is L. sakei in the inoculated samples over time?) and the levels of Listeria monocytogenes (stated as ‘the most worrying microorganism’) are required in order to complete the story of this manuscript. Also, despite some improvements to the language, further corrections are still required. The reviewer would still recommend the manuscript be submitted to another, most scientifically appropriate mdpi journal, such as Foods or Fermentation.

Response: The LAB levels in both batches (Control batch and Ls batch) are shown in lines L112-L115. In addition, the implantation of L. sakei 205, was tested at the last sampling time (90 days) in inoculated L. sakei batch. For this, 50% of the characteristics LAB colonies of MRS plates were randomly tested by sequencing analysis of 16S rRNA, and PFGE analysis by using the restriction NotI and SgsI enzymes (Thermo Fisher Scientific, USA) [Martín et al. (2021). Foods 2021, 10]. Most of the 85% of the investigated LAB isolates were identified as L. sakei by sequencing analysis showing these isolated the same pattern as L. sakei 205 in the PFGE analysis, thus, we can ensure that the L. sakei 205 strain was well-implanted in the dry-cured fermented sausages throughout their ripening. This paragraph has been included in the revised version of the manuscript. Regarding the effect of L. sakei 205 on reducing counts of L. monocytogenes, this has been demonstrated in previous work and it is not the target objective of this study, reason why we did not include this information in this research. In addition, as suggested by Reviewer 1, English has been revised and corrected. Finally, although we are extremely respectful of the opinion of Reviewer 1 about the suitability of the manuscript for the journal, we reckon our manuscript fits with the scope of Biology.

Reviewer 5 Report

The revised manuscript has been much improved. However, there are still some issues need to be addressed before consideration.

1, No substantive change was found in the revised manuscript for the comment "L 52, please provide the detailed information of in vivo assays. What are the research subjects?". Please explain.

2,  The texture of ripened “salchichón” should be placed after the subtitle of 'Volatile compound analysis' in Materials and Methods and Results and Discussion sections.

Author Response

The revised manuscript has been much improved. However, there are still some issues need to be addressed before consideration.

Response: We thank Reviewer 5 for the positive comments on the manuscript.

  1. No substantive change was found in the revised manuscript for the comment "L 52, please provide the detailed information of in vivo assays. What are the research subjects?". Please explain.

Response: Sorry for this. When we referred to “in vivo” assays, we meant that we tested the anti-L. monocytogenes activity of the lactic-acid bacteria in real “salchichón” samples under real environmental conditions occurring during their processing. This has been corrected in the revised version of the manuscript.

  1. The texture of ripened “salchichón” should be placed after the subtitle of 'Volatile compound analysis' in Materials and Methods and Results and Discussion sections.

Response: As suggested by Reviewer 5, the texture and color of ripened “salchichón” subtitles have been placed after those of the "Volatile compounds analysis" in both sections (Material and Methods and Results and Discussion).

Round 3

Reviewer 1 Report

The reviewer appreciates the efforts of the authors and the inclusion of the analysis of L. sakei (if only at day 90), however, this data (if only added as supplementary information) should be provided. The manuscript in general still requires a significant degree of language editing, especially in the newly added text.

Some of the data interpretation still also requires reviewing. 

Line 251 - From the table most of these values actually increase again at 90 days. Remarkable is quite an overstatement. 

Line 253 - 2-octanone levels are fairly stable on the LS batch while octanoic acid levels drop after 90 days. The reviewer also notes the exceedingly large error values for octanoic acid, which make the validity of these values questionable. 

Line 263 - Hexanal is greatly increased in the LS batch from 60 to 90 days.

Line 362 - The reviewer is not sure why this is remarkable as you have described an explanation of why it possibly occurs directly above.

I believe if the language is corrected and the aforementioned data interpretation better reviewed/worded this manuscript could be suitable for publication.

Author Response

The reviewer appreciates the efforts of the authors and the inclusion of the analysis of L. sakei (if only at day 90), however, this data (if only added as supplementary information) should be provided.

Response: We thank Reviewer 1 for the positive comment on the manuscript. We agree with Reviewer 1 that it is necessary to know if the inoculated L. sakei 205 strain is implanted throughout the ripening of “salchichón” being the predominant strain among LAB. This is demonstrated by the given information in the previously revised manuscript (“the implantation of L. sakei 205, was tested at the last sampling time (90 days) in inoculated L. sakei batch. For this, 50% of the characteristic LAB colonies of MRS plates were randomly tested by sequencing analysis of 16S rRNA, and PFGE analysis by using the restriction NotI and SgsI enzymes (Thermo Fisher Scientific, USA) [7]. Most of the 85% of the investigated LAB isolates were identified as L. sakei by sequencing analysis showing these isolates the same pattern as L. sakei 205 in the PFGE analysis. Thus, L. sakei 205 was well-implanted in the dry-cured fermented sausages throughout the ripening”). In our opinion, no additional information is necessary because the analysis of L. sakei is not the objective of this study.

The manuscript in general still requires a significant degree of language editing, especially in the newly added text.

Response: The language of the manuscript has been revised again.

Some of the data interpretation still also requires reviewing.

Line 251- From the table most of these values actually increase again at 90 days. Remarkable is quite an overstatement.

Response: We agree with Reviewer 1 that remarkable is quite an overstatement. This sentence has been rewritten in the new revised version of the manuscript and the word “remarkable” has been deleted.

Line 253- 2-octanone levels are fairly stable on the LS batch while octanoic acid levels drop after 90 days. The reviewer also notes the exceedingly large error values for octanoic acid, which make the validity of these values questionable.

Response: According to Reviewer 1 observation, this paragraph has been rewritten in the revised version of the manuscript as follows “…There was a significant (p ≤ 0.05) increase of 2-octanone in Control batch during ripening, while remained stable in Ls batch”.

In octanoic acid, after revision of the statistical analysis only differences (p ≤ 0.05) were found at day 90 in both Ls and Control batches. This has been amended in the revised version of the manuscript.

Line 263- Hexanal is greatly increased in the LS batch from 60 to 90 days.

Response: According to Reviewer 1 observation, the hexanal mean levels of Ls batch are higher on day 90 than on day 60 of ripening, but there are no significant (p ≤ 0.05) differences between these two days. Thus, we cannot say that there is a greatly increased in this compound from 60 to 90 days in this batch. In the revised manuscript the sentence “…a significant decrease of hexanal throughout the ripening time” has been modified by “…a significant decrease of hexanal for most of the ripening time”.

Line 362- The reviewer is not sure why this is remarkable as you have described an explanation of why it possibly occurs directly above.

Response: We agree with Reviewer 1 comment. The word “remarkable” has been deleted in this sentence.

I believe if the language is corrected, and the afore mentioned data interpretation better reviewed/worded this manuscript could be suitable for publication.

Response: We thank Reviewer 1 for his/her comments. All the above comments and suggestions have been attended to in the revised version of the manuscript.